# First Report of *Enterocytozoon hepatopenaei* Infection in Giant Freshwater Prawn (*Macrobrachium rosenbergii* de Man) Cultured in the Republic of Korea

**DOI:** 10.3390/ani12223149

**Published:** 2022-11-15

**Authors:** Gwang-Il Jang, Su-Mi Kim, Yun-Kyeong Oh, Soon-Jeong Lee, Sung-Youl Hong, Hyo-Eun Lee, Mun-Gyeong Kwon, Bo-Seong Kim

**Affiliations:** 1Aquatic Disease Control Division, National Fishery Products Quality Management Service, Busan 46083, Republic of Korea; 2Department of Aquatic Life Medicine, College of Ocean Science and Technology, Kunsan National University, Gunsan 54150, Republic of Korea

**Keywords:** *Enterocytozoon hepatopenaei*, EHP, giant freshwater prawn, HPM, monitoring

## Abstract

**Simple Summary:**

To our knowledge, this is the first reported detection of *Enterocytozoon hepatopenaei* (EHP) in cultured giant freshwater prawns (*Macrobrachium rosenbergii*) in 2021. While the prawn farms exhibited no clinical signs of EHP or mortality, histopathological and molecular biological analyses detected EHP during disease monitoring. The prevalence of EHP in infected prawn farms ranged from 4.9% to 18.2%. This is the first case of EHP infection in giant freshwater prawns cultured in the Republic of Korea.

**Abstract:**

In the Republic of Korea, *Enterocytozoon hepatopenaei* (EHP) was first isolated from Pacific whiteleg shrimp in April 2020; however, there are no existing reports of EHP infection in other shrimp or prawns. Here, we aimed to investigate EHP infection and its prevalence in giant freshwater prawn farms in the Republic of Korea. We tested prawns from 22 farms for EHP infection, and samples from eight farms showed positive EHP infection results in 2021. In EHP-infected prawn farms, the prevalence ranged from 4.9% to 18.2%. The prevalence of EHP infection in the Republic of Korea, derived from the prevalence in prawn farms, was estimated to be 0.8% in 2021. The proliferation of EHP was observed within the hepatopancreatic epithelial cells of prawns using H&E and Giemsa staining. Mature EHP was observed in the sinus between epithelial cells of the digestive tubules. Phylogenetic analysis revealed a clade distinct from the previously reported EHP in Pacific whiteleg shrimps. This is the first report of EHP infection in a giant freshwater prawn in the Republic of Korea, where the prevalence of EHP infection is not high, but it is recognized as an emerging disease that requires periodic monitoring and quarantine management in giant freshwater prawns.

## 1. Introduction

Hepatopancreatic microsporidiosis (HPM) is known as shrimp disease caused by infection due to *Enterocytozoon hepatopenaei* (EHP) with intracellular parasitic capacity. HPM is a wasting disease that results in economic loss due to growth retardation and reduced production [1]. EHP infects hepatopancreas epithelial cells, causing lesions with cell necrosis and rupture. This leads to functional impairment of nutrient storage, which is associated with growth performance [2,3], and energy metabolism, associated with growth hormone disorder [4]. EHP infection in shrimp farms can be induced by horizontal transmission through breeding seawater [5] and feces [6], direct transmission via cannibalism [7], feeding transmission with EHP-infected *Artemia salina* [8], and spreading transmission from surrounding residents such as polychaetes [9,10] and mussels [11]. Black tiger shrimp (*Penaeus monodon*), Pacific whiteleg shrimp (*Litopenaeus vannamei*), and blue shrimp (*Litopenaeus stylirostris*) were deemed susceptible host species against HPM [1,12,13]. Vietnam, India, Brunei, China, Indonesia, Malaysia, Venezuela, Australia, and the Republic of Korea were reported as countries where HPM infection most commonly occurred [13,14,15,16,17,18,19]. In the Republic of Korea, EHP was first isolated from Pacific whiteleg shrimp in April 2020 [18] and has been reported in multiple instances ever since [19]. However, there are no reports of EHP infection in other shrimp or prawns in the Republic of Korea.

The giant freshwater prawn *Macrobrachium rosenbergii* is a rapidly growing species under high temperature in subtropical and tropical regions [20]. *M. rosenbergii* is cultivated mainly in southern and southeastern Asian countries, including China, India, Thailand, and Taiwan [21]. Although *M. rosenbergii* can tolerate a wide range of temperatures (15–35 °C) and salinity (0–25 ppt) [22,23], the optimal temperature, salinity, and pH ranges for growth are 29–31 °C, 0–15 ppt, and 7.0–8.5, respectively [20,23,24]. The total global production of cultured *M. rosenbergii* significantly increased from 5000 tons in 1984 to 200,000 tons in 2002 [25], and tended to remain steady at 219,847 tons up till 2020 [26]. *M. rosenbergii* does not naturally inhabit the Republic of Korea, and its seeds are imported from abroad for domestic farming and small-scale farming. Various studies, including those involving monosex culture [27], integrated farming of rice and prawns [28], and feeding with probiotics [29], have been conducted to increase aquaculture production. In the Republic of Korea, 44 farms cultivate giant freshwater prawns, and integrated farming of rice and prawns is the most common aquaculture method. *M. rosenbergii* is a potential new species in the freshwater aquaculture industry of the Republic of Korea [30]. The National Fishery Products Quality Management Service (NFQS) regularly monitors EHP infection for disease control and prevention, and EHP was detected in eight farms in 2021. In this study, we aimed to investigate EHP infection in giant freshwater prawns in the Republic of Korea via histopathological and molecular biological analyses.

## 2. Materials and Methods

### 2.1. Sampling

Giant freshwater prawn farms were randomly selected, and 22 farms were sampled between April and November 2021. Twelve farms (four in Gyeongsangnam-do, seven in Gyeongsangbuk-do, and one in Ulsan) were sampled in April 2021, and ten farms (two in Gyeongsangnam-do, six in Gyeongsangbuk-do, one in Jeollanam-do, and one in Jeollabuk-do) were sampled from September to November 2021. No growth retardation or white feces associated with clinical signs of EHP infection were observed. From each farm, we sampled 29–30 prawns for molecular biological analysis and five prawns for histopathological analysis.

### 2.2. PCR Detection and Prevalence of EHP

The hepatopancreas samples extracted from 29–30 prawns were pooled into 6 lots of 5 shrimps each for DNA extraction using a 5 min DNA/RNA extraction kit (BioFactories, Monrovia, CA, USA) according to the manufacturer’s instructions. Extracted DNA was used to amplify the small subunit ribosomal RNA (*ssu rRNA*) gene of EHP using the nested PCR assay [7]. The *ssu rRNA* gene sequence of the NFQS_MR_EHP1 was compared with those in the GenBank databases using BLASTN [31]. The accession number of *ssu rRNA* gene sequence is OP363710. Phylogenetic analysis was performed using the MEGA X program [32]. A phylogenetic tree using a Jukes–Cantor (JC) model and uniform rates among sites based on the neighbor-joining method [33] was constructed using bootstrap analysis of 1000 replications.

The prevalence of EHP infection in prawn farms was measured from prevalence package including pooled samples analysis in R software using PCR results with test sensitivity and specificity of 95–100% and 90–95%, respectively, as described in a previous study [18].

### 2.3. Histopathological Analysis

For histopathological analysis, the hepatopancreases of giant freshwater prawns were separated and fixed in Davidson’s solution, dehydrated, embedded in paraffin wax, sectioned at 4 µm, and stained with hematoxylin and eosin (H&E) and Giemsa solution [18].

## 3. Results

### 3.1. Detection and Prevalence of EHP Infection

As shown in Table 1, EHP was detected in eight giant freshwater prawn farms between April and November 2021. Positive PCR results were obtained for four out of six pooled samples. The prevalence of EHP infection within each prawn farm ranged from 4.9% (95% confidence interval: 0.2–15.0%) to 18.2% (95% confidence interval: 2.3–40.2%) using PCR results (Table 1). The overall incidence of EHP infection among giant freshwater prawn farms in the Republic of Korea from April to November 2021 was estimated to be 0.8% (95% confidence interval: 0–2.3%).

### 3.2. Sequence Analysis Using Nested PCR

The *ssu rRNA* gene sequence NFQS_MR_EHP1 was identical (100% similarity) to seven other sequences obtained in this study (Figure 1). BLAST analysis revealed that the *ssu rRNA* gene sequence similarities between NFQS-EHP1 and all reference sequences isolated from India, Vietnam, China, and Thailand [acquired from the GenBank database http://www.ncbi.Nlm.nih.gov/genbank (accessed on 5 October 2022)] ranged from 99.6% to 99.7%.

The phylogenetic tree based on the *ssu rRNA* gene sequences revealed that the EHP sequence obtained from the giant freshwater prawn (designated NFQS_MR_EHP1) was clearly distinct from the previously reported EHP sequences and formed a monophyletic clade (Figure 1).

### 3.3. Histopathological Observation of Giant Freshwater Prawn Macrobrachium rosenbergii

Plasmodium clumps, formed due to the proliferative activity of EHP, were observed histopathologically inside epithelial cells in the hepatopancreas (Figure 2A). These lesions were more pronounced with Giemsa staining than with H&E staining (Figure 2C). Although the H&E-stained sinus between hepatopancreas epithelial cells was filled with an eosinophilic fluid containing a large amount of protein (Figure 2B), multiple mature EHP cells ranging from 0.8 to 1 µm were observed in the Giemsa-stained sinus (Figure 2D). There were no other lesions associated with inflammation, such as hemocytic infiltration or serous exudate accumulation.

## 4. Discussion

During monitoring for effective disease control and prevention, EHP was identified in clinically healthy giant freshwater prawns using histopathological and molecular biological analyses. In the Republic of Korea, since the first report of EHP infection in Pacific whiteleg shrimp in Ganghwa-gun in 2020 [18], EHP has been detected in various regions [19].

In the phylogenetic tree constructed from the giant freshwater prawn sequences, NFQS_MR_EHP1 gene sequence showed 99.6–99.7% identity to the EHP isolates from India, Vietnam, China, and Thailand. The NFQS_MR_EHP1 gene sequence obtained in this study formed a distinct branch with NFQS-EHP1 (MZ819965) sequence detected from Pacific whiteleg shrimp that was previously reported in 2021 in the Republic of Korea. Artanto et al. (2019) also found that the *ssu rRNA* gene of EHP in Indonesia was highly homologous with the *ssu rRNA* genes of EHP detected in India and China [34].

EHP ranging from 0.8 to 1 µm were observed in epithelial cells and sinuses of the hepatopancreas in giant freshwater prawns; other lesions, including regressive and inflammatory changes, such as necrosis of epithelial cells, hemocytic infiltration, and serous exudate accumulation in the hepatopancreas, were absent [18,35]. The absence of lesions associated with organ malfunction might be due to insufficient EHP infection to cause clinical or histopathological changes. Other studies on EHP infection in giant freshwater prawns have reported growth retardation [36] and various lesions, including necrotic and collapsed epithelial cells of the hepatopancreas; inflammation with hemocytic infiltration; disappearance of B-, R-, and F-cells; and increased karyomegaly [37]. In this study, EHP was detected during monitoring of pathogen retention in clinically healthy prawns, in which it can be difficult to detect symptoms or lesions. This is distinct from cases of sudden mortality or growth retardation, wherein symptoms or lesions are prominent. The health of prawns infected with EHP can deteriorate with an increase in ammonia and nitrite contents [38,39], leading to complex infections related to bacterial diseases that may cause mortality [40,41].

In this study, the incidence of EHP infection in prawn farms was relatively low, ranging from 4.9 to 18.2%, in contrast to incidence ranging from 30 to 93% in other shrimp farms [40,42,43]. In this study, the low prevalence and mild EHP infection were consistent with the absence of white feces caused by a complex infection of *Vibrio* sp. and EHP [44]. Aquaculture regimes for giant freshwater prawns in the Republic of Korea involve indoor culture for 2 months and integrated farming in rice fields for 5 months [30]. Accordingly, prawn farmers may find it difficult to monitor growth and report the occurrence of EHP infection prior to the harvest process. Although EHP disease is more infectious under high-salinity (> 15 ppt) than under low-salinity conditions (<5 ppt), caution regarding EHP infection is needed in giant freshwater prawns because EHP infection can be transmitted at salinity ranging from 3 to 30 ppt [45]. Giant freshwater prawn infection can easily occur because of the salinity, ranging from 0 to 12 ppt, of the prawn farming environment in the Republic of Korea [30].

The lack of any drug treatment for HPM [8,16], transmission of various surrounding organisms [8,9,11], and spread in several regions [19] make it difficult to effectively treat the disease upon occurrence. Therefore, periodic monitoring should be performed preemptively to control and prevent EHP transmission. This study brings about the first reported case of EHP infection in giant freshwater prawn farms in the Republic of Korea.

## 5. Conclusions

In conclusion, this is the first report of EHP infection in giant freshwater prawns (*M. rosenbergii*) in the Republic of Korea. EHP was detected in eight farms between April and November 2021, and the prevalence of HPM within each farm ranged from 4.9% to 18.2% using PCR results. EHP was detected histopathologically in epithelial cells and sinuses of hepatopancreas without inflammation, such as hemocytic infiltration. Hence, it is crucial to monitor and control the status of HPM in *M. rosenbergii* in the Republic of Korea.

## Figures and Tables

**Figure 1 animals-12-03149-f001:**
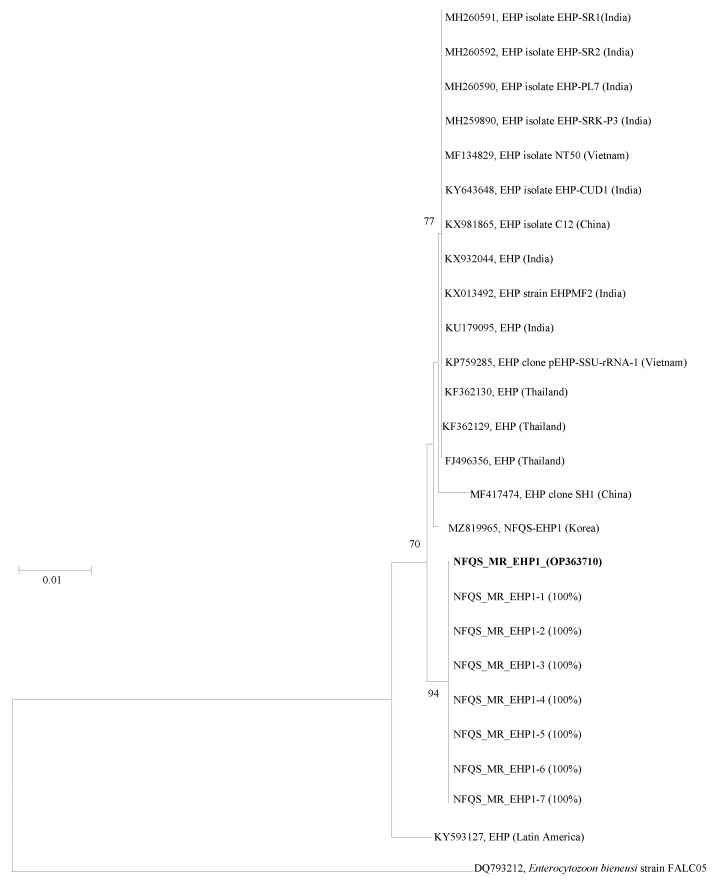
The phylogenetic tree was constructed by the neighbor-joining method, using MEGA software (version X; http://www.megasoftware.net (accessed on 5 October 2022)). All reference sequences were acquired from the GenBank database (http://www.ncbi.nlm.nih.gov/genbank (accessed on 5 October 2022)). *Enterocytozoon bieneusi* (DQ793212) was used as the outgroup. Only bootstrap values above 70% are shown (1000 resamplings) at branch points. Bar, 0.01 substitutions per site. Bold refers to a strain separated from giant freshwater prawn in Korea.

**Figure 2 animals-12-03149-f002:**
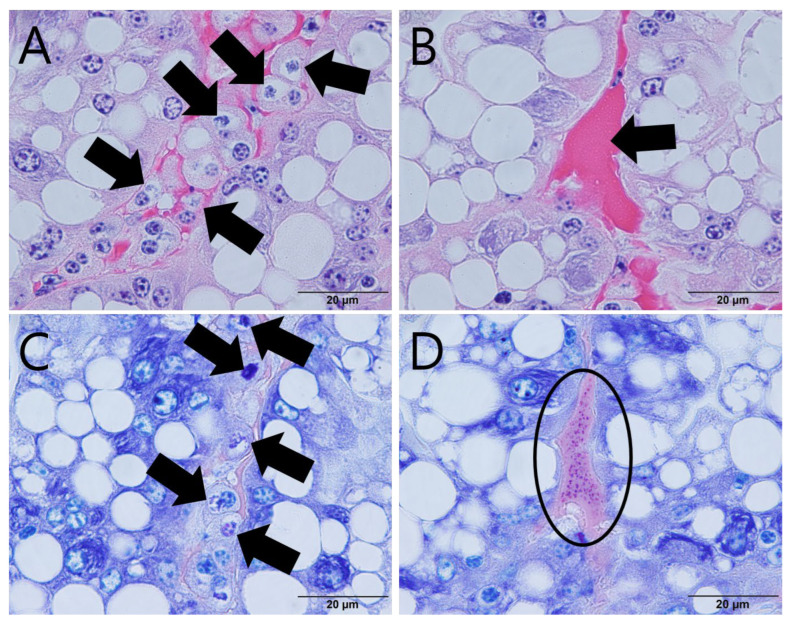
Hepatopancreas in giant freshwater prawn (*Macrobrachium rosenbergii*) infected with EHP. H&E staining revealed: (**A**) plasmodium of EHP in epithelial cells of hepatopancreas (arrows) and (**B**) sinus between epithelial cells of digestive tubule filled with eosinophilic fluid (arrow). Giemsa staining revealed: (**C**) plasmodium of EHP in epithelial cells of hepatopancreas (arrows) and (**D**) multiple mature EHPs ranging from 0.8 to 1 μm in sinus (ellipse).

**Table 1 animals-12-03149-t001:** Prevalence estimation of EHP infection with PCR test results of giant freshwater prawn *Macrobrachium rosenbergii*. PCR test condition set as 95–100% sensitivity and 90–95% specificity for prevalence estimation.

Sampling Date	Province	Sampling Site	Length (Mean ± SD, cm)	No. of Specimen	Detection Numbers of Pooled Sample/Test Numbers of Pooled Samples	Prevalence of EHP Infection in Shrimp Farm (Average Prevalence /95% Confidence Interval)
April 2021	Gyeonsangnam-do	Geoje-si	6.7 ± 1.3	30	1/6	4.9% (0.2–15.0%)
		Sancheong-gun	13.3 ± 1.6	30	1/6	4.9% (0.2–15.0%)
		Changnyeong-gun	8.6 ± 1.3	29	0/6	-
		Hadong-gun	6.2 ± 0.6	29	0/6	-
	Gyeonsangbuk-do	Gyeongju-si	5.2 ± 0.9	30	2/6	7.3% (0.3–21.3%)
		Goryeong-gun	3.4 ± 0.9	30	0/6	-
		Mungyeong-si	7.5 ± 1.0	30	1/6	4.9% (0.2–15.0%)
		Yeongyang-gun	7.9 ± 0.9	30	3/6	11.3% (0.6–28.6%)
		Yeongcheon-si	7.9 ± 2.0	30	0/6	-
		Cheongsong-gun	6.7 ± 1.0	30	1/6	4.9% (0.2–15.0%)
		Chilgok-gun	8.4 ± 1.2	30	0/6	-
	Ulsan	Ulju-gun	N.D.	30	0/6	-
September 2021	Gyeonsangnam-do	Sancheong-gun	7.9 ± 1.6	30	0/6	-
October 2021	Gyeonsangbuk-do	Gyeongju-si	6.1 ± 0.7	30	2/6	7.3% (0.3–21.3%)
		Gimcheon-si	11.2 ± 1.7	30	0/6	-
		Mungyeong-si	11.2 ± 1.4	30	0/6	-
		Yeongyang-gun	10.0 ± 1.3	30	0/6	-
		Cheongsong-gun	10.2 ± 2.0	30	4/6	18.2% (2.3%–40.2%)
		Chilgok-gun	11.5 ± 2.1	30	0/6	-
	Jeollanam-do	Hampyeong-gun	12.9 ± 2.3	30	0/6	-
	Jeollabuk-do	Wanju-gun	7.5 ± 1.6	30	0/6	-
November 2021	Gyeonsangnam-do	Hadong-gun	8.2 ± 0.9	30	0/6	-
Total	22 times of sampling			658	15/132	0.8% (0–2.3%)

N.D.: not determined.

## Data Availability

All data are contained within the article. Please contact the corresponding author for additional data requests.

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
