# Peer review of "First Report of Enterocytozoon hepatopenaei Infection in Giant Freshwater Prawn (Macrobrachium rosenbergii de Man) Cultured in the Republic of Korea"

_animals, 2022, doi:10.3390/ani12223149_

Round 1

Reviewer 1 Report

Review of ‘First Report of Enterocytozoon hepatopenaei Infection in Giant Freshwater Prawn (Macrobrachium rosenbergii de Man) Cultured in the Republic of Korea by Gwang-Il Jang, Su-Mi Kim, Yun-Kyeong Oh, Soon-Jeong Lee, Sung-Youl Hong, Hyo-Eun Lee, Mun-Gyeong Kwon and Bo-Seong Kim.

The authors studied farmed for pathogens and detected, for the first time, the presence of Enterocytozoon hepatopenaei infection. They provided the results of PCR and histopathological observations. The authors discussed the results and concluded that there is a need to monitor giant freshwater prawn farms for this new pathogen.

 After some minor revisions, this paper can be recommended for publication in ANIMALS.

General remark.

Pg 2 Ln 57: The authors should include more recent data on the production of Macrobrachium rosenbergii

Pg 2 Ln 72-75: Please, insert a map of the study area or provide coordinates for each location mentioned.

Table 1. Please, provide standard deviations for the mean body lengths

Pg 6 Ln 183-186: The authors discussed the possible role of salinity as a factor promoting EHP infection, but presented no information about environmental conditions in the giant freshwater prawn farms.

References: The Latin names should be italicized.

Pg 2 Ln 57: Suggest changing ‘has significantly increased’ to ‘significantly increased’

Pg 3 Ln 99: Suggest changing ‘the hepatopancreas’ to ‘the hepatopancreases’

Author Response

we attached the file associated with reply to the suggestions.

we would like to express our sincere gratitude towards the reviewers for their positive and constructive criticism to our manuscript. The manuscript has vastly benefited from your valuable and insightful comments and suggestions. We look forward to hearing from you and would be happy to address any further concerns, if required.

Reviewer 2 Report

The manuscript describes the first finding of Enterocytozoon hepatopenaei in Macrobrachium rosenbergii in Korea. The pathogen is found by PCR screening of randomly selected farms, and the positive findings are supported by histology and DNA sequencing.

The manuscript is very well written, and the findings appear to be solid. Below I have listed some minor issues I have come across. My only serious critique would be that the findings do not provide any new biological insight into E. hepatopenaei or the disease caused by it, but is merely a report on the finding of E. hepatopenaei in M. rosenbergii in seemingly symptomless animals, and an estimation of the prevalence of the pathogen in farms in Korea. I am therefore not sure that the manuscript will be interesting enough for a broad audience in order to merit its publication in a journal like Animals, but that will up to the editor to decide.

Throughout the paper, the term “shrimp” is used for Penaeid shrimps, while the term “prawn” is used for M. rosenbergii. I don’t think there is consensus about which species to call shrimps, and which to call prawns, and the two terms are often used interchangeably. I thus recommends to either start defining what is meant with these terms in the beginning of the manuscript, or stick to e.g. “penaeid shrimp” and “M. rosenbergii” or whatever makes sense in the specific context.

Line 29: It is stated that this study is the first to report EHP infection in giant freshwater prawns, but it needs to be added that this only holds for Korea, as it has been found in this species before in other countries.

Line 51: Is it the animals that are “rapidly growing” or the farming industry?

Line 92 – 94: More details about parameters used for phylogenetic tree are needed.

Line 107 – 110: It is not evident how the prevalence values were calculated, given that pools of animals were tested. Please explain in more details.

Line 117 and 125: Should the reference not be to Figure 1 rather than 2?

Line 137-138: Is it stated that Figure 2 shows mature EHP of a size of 0.8 – 1 um. However, when comparing to the scale bar of the pictures, the particles in question looks considerably smaller than 1 um. Please explain.

Author Response

(The authors gave the same response as above.)

Reviewer 3 Report

Can you have more results from the molecular analysis presented?

Author Response

Thanks for your suggestion. we provided all data of molecular analysis in manuscript. 

we would like to express our sincere gratitude towards the reviewers for their positive and constructive criticism to our manuscript. The manuscript has vastly benefited from your valuable and insightful comments and suggestions. We look forward to hearing from you and would be happy to address any further concerns, if required.